# Impact of Extubation Time on Feeding Outcomes after Neonatal Cardiac Surgery: A Single-Center Study

**DOI:** 10.3390/children10030592

**Published:** 2023-03-20

**Authors:** Jeffrey W. Kepple, Meghan Kendall, Laura A. Ortmann

**Affiliations:** 1Department of Pediatrics, Creighton University School of Medicine, 2500 California Plaza, Omaha, NE 68178, USA; 2Department of Pediatrics, University of Nebraska Medical Center, 42nd and Emile, Omaha, NE 68198, USA

**Keywords:** congenital heart surgery, neonate, pediatrics, feeding

## Abstract

This study aimed to examine the impact of timing of extubation on feeding outcomes in neonates after surgery for congenital heart disease. This was a single-center retrospective study between December 2014 and June 2020. Patients were divided into three categories: extubated in the OR (immediate), extubated in the intensive care unit (ICU) between 0 and 3 days post-procedure (early), and extubated >3 days post-procedure (delayed). Comparing the immediate and early groups, we found no difference in time to first enteral feed (1.3 days (1.0–3.4) vs. 2.3 days (1.1–3.3), *p* = 0.27). There was no difference in time to first oral feed (2.0 days (1.1–4.5) vs. 3.1 days (1.8–4.4), *p* = 0.34) and time to goal feed (6.0 days (3.2–8.3) vs. 6.9 days (5.0–9.0), *p* = 0.15)). There was no difference in all oral feeds at one year: 88% vs. 98%, *p* = 0.16. The delayed extubation group performed significantly worse on all measures. Immediate and early extubation displayed no differences in feeding outcomes and length of stay in this study, while delayed extubation performed worse on all measures. Thus, we believe that clinicians should emphasize extubation within 3 days post-surgery to improve feeding outcomes while minimizing time hospitalized.

## 1. Introduction

Recently, there has been an increase in the rate of children being extubated in the operating room (OR) after surgery for congenital heart disease (immediate extubation) in hopes of shortening overall hospital stay [1]. The published studies on the risks and benefits of immediate extubation have yielded varying results. Some studies have shown that early extubation after cardiac surgery is safe for most patients, including neonates, without increased reintubation rates [2,3,4,5,6]. However, there is a lack of consensus whether hospital length of stay is decreased [5,7,8]. In pediatric cardiac surgery patients, the first time extubation success rate is between 12 and 19%, with differences seen depending on cardiac pathology and operation [3,9]. Specific risk factors that decrease the likelihood of successful extubation include younger age, lower weight, longer time on cardiopulmonary bypass, and lung diseases that result in airway edema or pulmonary hypertension [1,3,4,7,10,11].

Neonates who undergo surgery for congenital heart disease have well documented difficulty reaching feeding goals. These include increased time to goal feeds and prolonged transition to oral feeds [12]. A study from 2007 showed that 11% of patients had prolonged time to reach goal feeds, while 45% had prolonged time to transition to oral feeds and thus required gastrostomy tube placement prior to discharge [12]. Use of cardiopulmonary bypass in cyanotic congenital heart disease may also impact initiation of gavage feeds, first nipple feeds, and maximal gavage feeds [13]. The difficulties surrounding feeding patterns result in longer hospital stays, increased costs to the patient, and overall poor outcomes.

While there have been shown to be benefits associated with early extubation in neonates undergoing cardiopulmonary bypass, there has been limited research exploring the feeding patterns of patients who were extubated in the operating room compared to those extubated later. It has been previously shown that longer intubation times result in delayed feeding [14], but these data do not compare the impact of immediate extubation to early extubation [2]. The goal of this study was to assess the impact of immediate extubation in the OR on feeding success in neonates after surgery for congenital heart disease.

## 2. Methods

This was a single-center retrospective cohort study of neonates <30 days of age who had cardiac surgery between December 2014 and June 2020. This study was approved by the institutional review board and the requirement for informed consent was waived. Exclusion criteria included upper airway or gastrointestinal defects preventing oral feeds; tracheostomy placement pre-operatively or within 4 weeks post-operative; diagnosis of trisomy 13, 18, or 21; death during initial hospital stay; or if multiple cardiac procedures were performed within 30 days. Children undergoing isolated PDA ligation were also excluded. Demographic data collected included age and weight at the time of surgery, cardiac diagnosis and surgery, and pre-operative feeding regiment.

Patients were divided into three groups for analysis: (1) extubated in the OR (immediate), (2) extubated in the intensive care unit (ICU) between 0 and 3 days post procedure (early), (3) and extubated >3 days post procedure (delayed). The primary outcome was time to reach the goal volume of feeds. Feeding goals were determined for each patient individually by a dedicated cardiac ICU dietician. Secondary outcomes included ICU and hospital length of stay, method of feeding at discharge, and ability to exclusively orally feed at 1 year of age.

Categorical data were reported as number (percent) and continuous data as medians (interquartile range). Differences in demographics and outcomes were analyzed using the Kruskal–Wallis test for three-way comparisons and Students t-test or Fisher’s test for two-way comparisons. Statistical significance was set at *p* < 0.05. Pearson’s correlation coefficient was used to determine if baseline continuous variables were correlated with the primary outcome: time to goal feeds. Data were analyzed using SAS Studio 3.8 (SAS Institute, Care, NC, USA).

## 3. Results

In total, 119 patients were included in the analysis, 32 (27%) with immediate extubation in the OR, 43 (36%) with early extubation, and 44 (37%) with late extubation. There were minimal demographic differences between immediate and early extubation patients (Table 1). Late extubation patients were more likely to have cyanotic heart disease and have more complex operations with longer cardiopulmonary bypass and cross clamp times.

Feeding outcomes between groups are listed in Table 2. When comparing the immediate and early extubation groups, there was no difference in time to goal feeds (6.0 days (3.2–8.3) vs. 6.9 days (5.0–9.0), *p* = 0.15)) (Figure 1). There was also no difference in time to first enteral feed (1.3 days (1.0–3.4) vs. 2.3 days (1.1–3.3), *p* = 0.27) and time to first oral feed (2.0 days (1.1–4.5) vs. 3.1 days (1.8–4.4), *p* = 0.34). There was no difference in patients receiving all oral feeds at one year: 88% vs. 98%, *p* = 0.16. The delayed extubation group performed significantly worse on all measures.

Enteral feeding pre-operatively was associated with a shorter time to goal feeds postoperatively (5.6 days (4.1–7.7) vs. 7.4 days (6.4–11.5), *p* = 0.036). Males reached goal feeds sooner than females (5.7 days (3.4–7.3) vs. 7.5 (5.9–9.0), *p* = 0.005). There was no correlation between age (r = 0.06), gestational age (r = −0.002), weight (r = −0.03), cardiopulmonary bypass time (r = 0.11), or cross clamp time (r = 0.04) and time to goal feeds. There was no association between time to goal feed volume and post-operative cyanosis (cyanotic 6.3 days (4.9–7.4) vs. non-cyanotic 6.3 days (4.9–7.4), *p* = 0.95).

When comparing the immediate and early groups, there was no difference in length of hospital stay (15.0 days (11.6–19.8) vs. 14.0 days (12.3–18.2), *p* = 0.30) (Figure 2).

## 4. Discussion

Our study of neonates undergoing surgery for congenital heart disease demonstrated no difference in feeding patterns and hospital length of stay in patients extubated in the operating room vs. within the first three post-operative days. This could have been due to a couple of different factors. First, these studies did not separate the patients into different groups after the immediate extubation group. Medically complex patients have more risk factors for prolonged intubation, such as pulmonary hypertension and prolonged time on bypass. These variables lead to increased time to extubation and likely higher rates of failed extubation [3,11]. For this reason, we split patients into three groups for the analysis in an attempt to isolate extubation timing’s impact on feeding patterns. Multiple studies have defined prolonged mechanical ventilation post-surgery as 72 h, and this was used as the distinguisher for this population [15,16,17]. With a lack of feeding outcome difference in immediate vs. early extubation, some patients may benefit from minimizing risk of immediate extubation while experiencing similar outcomes.

The goal of this study was to further examine the impact of immediate extubation after surgery for patients with congenital heart disease. Since there are varying practices present across hospitals [18], it is important to attempt to clarify which practices are best in this population. Past studies have compared the early extubation to delayed extubation groups, but further investigating the early extubation group was necessary to tease out optimal practice. While studies have shown that early extubation within 4–6 h is safe and has no increased risk for failed extubation [2,3,4,5,6,7], it remains unclear as to how beneficial this early extubation was to clinical outcomes such as length of stay and feeding outcomes [7]. We believe that this study, which differentiated the time period of extubation more clearly, allows for clinicians to see the benefit of extubation prior to 72 h post-operation without necessarily extubating within the first 6 h.

Children with delayed extubation likely had a higher severity of illness, as shown by longer cardiopulmonary bypass and cross clamp times, which will impact time to achieve feeding goals. While pediatric post-cardiac corrective surgery patients have been shown to be delayed in reaching feeding goals [12], these data showed that there may be nuance in determining feeding outcomes in these patients. Extubation prior to 72 h post-surgery resulted in earlier feeding goal achievement compared to prolonged intubation. While we observed increased NG tube use upon discharge in patients with delayed goal feeds, NG tubes are also associated with increased complications and readmissions [19]. Thus, optimizing extubation timing may prevent readmission and other feeding complications that commonly arise in this population.

There is limited data on the impact of pre-operative enteral feeding; in this study, pre-operative feeding was associated with shorter time to goal feeds in this study. It is possible that patients who were able to feed prior to their operation were healthier at baseline and thus were able to restart feeding quicker than those who were not fed. Another possible explanation is that neonates who have had pre-operative enteral feeding progressed in their diet more aggressively than those who did not due to provider bias. More investigation is needed to examine the impact of pre-op feeding on time to reaching goal feeds in the effort to have more favorable outcomes along with shorter length of stay in the hospital.

One interesting finding seen in this study was that males reached goal feed volume faster than females. The literature review reveals that there are no consistent differences in feeding patterns in male and females in the neonatal period. It is likely that this finding was simply due to the fact that the males present in the population analyzed had less severe pathology and thus did not require as much time prior to reach goal feeds.

While time to goal feed and length of stay are important quality measures, we wanted to ensure that longer-term feeding patterns were not impacted between the groups present. We saw no difference in oral feeding at 1 year of age between the immediate and early extubation groups. While not significant, there were slightly higher rates of oral feeds with the early extubation group compared to immediate extubation. This demonstrates that immediate extubation also did not have any long-term impact on these patients. Future studies should examine in detail the long-term impact of extubation timing and feeding patterns.

While a wide variety of cardiac diagnoses and operations were present in this study, we did not find any specific correlations with any outcome measured. This could have been due to the smaller population subsets in this study, or it is possible that these patients experienced similar feeding patterns, regardless of pathology or operation performed. We encourage future studies to examine this relationship so individualized care can be provided to each patient.

Limitations were inherently present in this study. First, this was a single-center study resulting in a lack of external reliability for other medical centers. Second, the determination of goal feeds was not standardized and could have been impacted by provider biases. Additionally, we did not include patients who received multiple operations within a 30-day period. Therefore, these data should not be generalized to all patients receiving cardiac corrective surgery, but we encourage further investigation of this subset of patients requiring multiple operations while hospitalized.

There are multiple avenues for further investigation into this topic. First, while prolonged intubation was defined as greater than 72 h similar to previous studies [15,16,17], we saw the mean time to extubation in the group prior to 72 h was 1.6 days compared to 5.2 days for the delayed extubation. This could point towards the utility of an earlier cutoff for improved outcomes in extubation for this population. Despite this nuance, we still recommend individualized decision making for extubation of each patient.

One other aspect that is important to consider in this setting is the sustainability of clinical changes with changing practices. Specifically, it has been shown that after implementation of the Pediatric Heart Network Collaborative Learning Study, the rates of early extubation within 6 h rose from 12% to 67% [10]. However, upon re-evaluation of the same hospital systems, it was shown that these changes were not carried out past the first year of implementation [20]. With this information, we believe it is important to distinguish the need for extubation within 3 days rather than focusing on extubation soon after ICU admission, as there may not be a difference in outcomes.

Immediate and early extubation displayed no differences in this study, and thus we believe that clinicians should emphasize extubation within 3 days post-surgery to improve feeding outcomes while minimizing time hospitalized. Further investigation is required to determine the impact of specific cardiac diagnosis as well as evidence from multiple medical centers on extubation and feeding patterns.

## Figures and Tables

**Figure 1 children-10-00592-f001:**
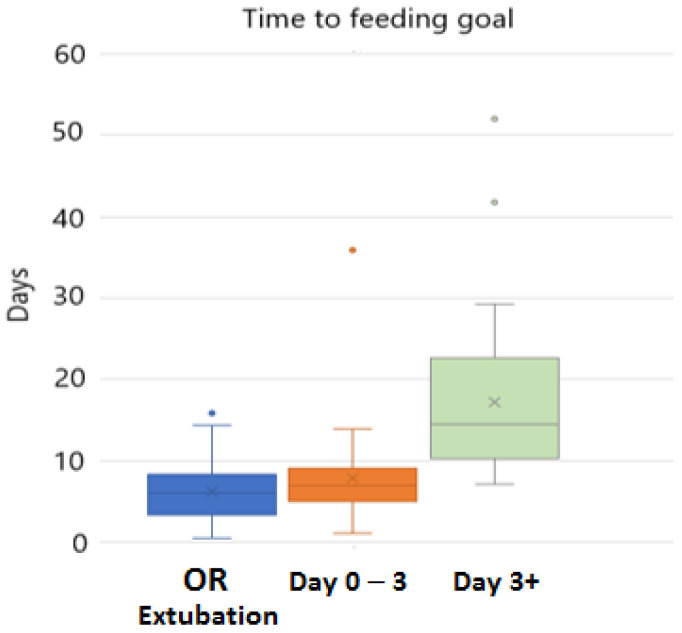
Time to feeding goal. OR = immediate extubation; error bars display IQR.

**Figure 2 children-10-00592-f002:**
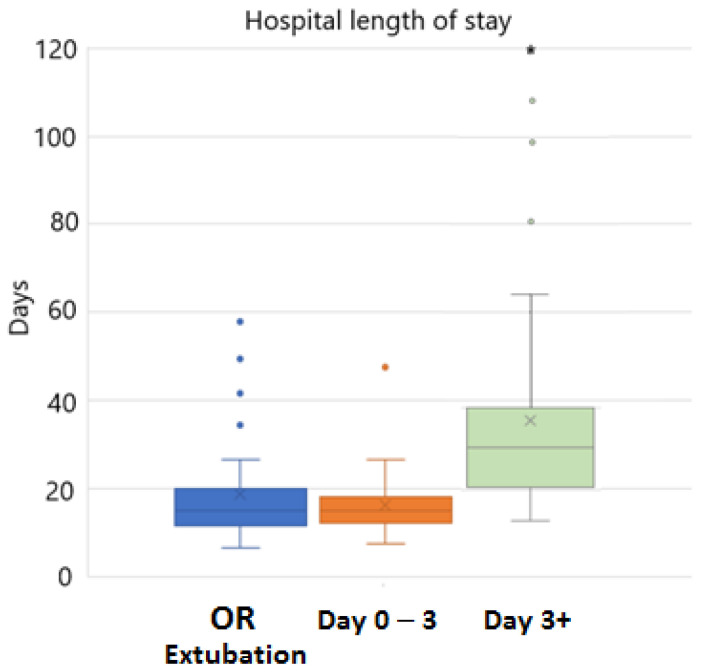
Hospital length of stay. OR = immediate extubation; error bars display IQR. * represents one outlier with hospital stay >120 days.

**Table 1 children-10-00592-t001:** Demographic and clinical characteristics based on timing of extubation.

Variable	Immediate Extubation *n* = 32	Early Extubation *n* = 43	Delayed Extubation *n* = 44	*p*-Value
Age, days	5.0 (3.0–6.5)	4.0 (3.0–6.0)	5.5 (4.0–7.0)	0.117
Gestational age	39.0 (37.6–39.2)	39.0 (38.0–39.2)	38.4 (37.3–39.0)	0.049
Weight, kg	3.3 (2.9–3.6)	3.3 (3.1–3.8)	3.2 (2.7–3.6)	0.121
Sex (male)	16 (50%)	31 (72%)	26 (59%)	0.14
Lesion				0.295
Aortic arch hypoplasia/coarctation	9 (27%)	13 (30%)	11 (25%)	
Hypoplastic left heart	4 (13%)	2 (5%)	9 (20%)	
Interrupted aortic arch	1 (3%)	1 (2%)	3 (7%)	
Other two ventricles	3 (9%)	2 (5%)	3 (7%)	
Other single ventricle	1 (3%)	2 (5%)	2 (5%)	
Pulmonary atresia/IVS ^1^	1 (3%)	4 (9%)	2 (5%)	
Pulmonary atresia/VSD ^2^	2 (6%)	0	3 (7%)	
TAPVR ^3^	2 (6%)	4 (9%)	4 (9%)	
TGA ^4^	9 (28%)	13 (30%)	3 (7%)	
Truncus arteriosus	0	2 (5%)	4 (9%)	
Pre-operative feeds				0.093
Oral	22 (69%)	23 (53%)	16 (36%)	
Nasogastric	3 (9%)	5 (12%)	12 (27%)	
Oral + nasogastric	0	1 (2%)	2 (5%)	
None	7 (22%)	14 (33%)	14 (32%)	
Post-operative cyanosis	14 (42%)	16 (37%)	31 (70%)	0.005
Surgery				0.058
Arch reconstruction	10 (31%)	11 (40%)	16 (36%)	
Arterial switch	8 (25%)	12 (28%)	2 (5%)	
BTT shunt	1 (3%)	4 (9%)	0	
Norwood + Sano	4 (13%)	2 (5%)	7 (16%)	
Other two ventricle repair	6 (19%)	2 (5%)	9 (20%)	
Other single-ventricle palliation	1 (3%)	1 (2%)	2 (5%)	
TAPVR repair	1 (3%)	4 (9%)	4 (9%)	
Truncus arteriosus repair	1 (3%)	1 (2%)	4 (9%)	
STAT ^5^ category				<0.0001
1	8 (26%)	1 (2%)	2 (5%)	
2	1 (3%)	2 (5%)	4 (10%)	
3	7 (23%)	11 (26%)	5 (12%)	
4	11 (36%)	27 (63%)	23 (55%)	
5	4 (13%)	2 (5%)	8 (19%)	
Cardiopulmonary bypass time, minutes	83 (53–119)	86 (61–100)	120 (83–155)	0.0004
Cross clamp time, minutes	39 (15–48)	36 (19–52)	58 (39–77)	0.0004

^1^ IVS: intact ventricular septum, ^2^ VSD: ventricular septal defect, ^3^ TAPVR: total anomalous pulmonary venous return, ^4^ TGA: transposition of the great arteries, ^5^ STAT: The Society of Thoracic Surgeons-European Association for Cardio-Thoracic Surgery (higher STAT category is associated with an increased risk of mortality).

**Table 2 children-10-00592-t002:** Clinical outcomes based on timing of extubation.

Variable	Immediate Extubation *n* = 32	Early Extubation *n* = 43	Delayed Extubation *n* = 44	*p*-Value
Time to extubation, days	0	1.6 (0.9–2.1)	5.2 (4.2–10.6)	<0.0001
Time to first enteral feed, days	1.3 (1–3.4)	2.3 (1.1–3.3)	3.5 (2.2–5.1)	0.0009
Time to first oral feed, days	2.0 (1.1–4.5)	3.1 (1.8–4.4)	8.4 (5.3–15.6)	<0.0001
Time to goal feeds, days	6.0 (3.2–8.3)	6.9 (5.0–9.0)	14.5 (10.4–22.3)	<0.0001
Length of hospital stay, days	15.1 (11.6–19.8)	15.0 (12.3–18.2)	32.5 (21.2–41.6)	<0.0001
z-score change	−0.8 (−1.1–−0.48)	−1.0 (−1.3–−0.47)	−1.2 (−1.5- −0.74)	0.057
Type of first feeds post-op				0.0001
Oral	17 (53%)	22 (51%)	5 (11%)	
Nasogastric bolus	9 (28%)	7 (16%)	13 (30%)	
Nasogastric continuous	6 (19%)	14 (33%)	26 (59%)	
Discharge feeds				0.007
All oral	21 (66%)	30 (70%)	15 (34%)	
Oral + tube	7 (22%)	9 (21%)	15 (34%)	
All tube	4 (12%)	4 (9%)	14 (32%)	
Feeds at one year				0.0002
All oral	24 (75%)	41 (95%)	27 (61%)	
Oral + tube	3 (9%)	1 (2%)	3 (7%)	
All tube	1 (3%)	0	11 (25%)	
Missing data	4 (13%)	2 (5%)	3 (7%)	

## Data Availability

Data available by request.

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
