# Peer review of "Impact of Extubation Time on Feeding Outcomes after Neonatal Cardiac Surgery: A Single-Center Study"

_children, 2023, doi:10.3390/children10030592_

Round 1
Reviewer 1 Report
A very interesting and educational manuscript that has clinical merit; however, there are some editing issues that the authors should consider and address. The following are my suggestions/comments regarding those issues. Line 21, "... of stay in this study, while delayed extubation did worse in all measures. Line 33, "... cardiac surgery patients, the first time ...". Line 48, "... there have been benefits associated ...". Line 50, "... extubated in the OR compared to ...". Line 51, "... intubation times resulted in delayed ...". Line 58, "the Institutional Review Board and the ...". Line 64, "... and pre-operative feeding regimens." Line 70, "... ability to exclusively be orally fed at ...". Line 75, "Statistical significance was set al ...". Line 80, "In total, 119 patients ...". Line 84, "... operations with longer times on cardiopulmonary bypass ...". Line 99, "... worse in all measures." Lines 116 & 117, "...extubated in the OR vs. within ...". Line 121, "These variables lead to increased ...". Line 122, "... for the analysis in an attempt to ...". Line 140, "longer times on cardiopulmonary bypass ...". Line 142, "... that there may be a nuance in ...". Line 149, "There is a limited day on the impact of pre-operative enteral feedings. In this study, pre-operative ...". Line 150, "... with shorter times to goal feeds. It is possible ...". Line 156, "in an effort to have more favorable ...". Line 158, "... reached goal feed volumes faster ....". Line 164 & 165, "... between the groups. We saw no difference ...". Line 179, "... of goal feeds was not standardized and ...". Line 201, "... specific cardiac diagnoses as well as evidence ...".
Author Response
Thank you for your closer reading of our manuscript and your comments. We have revised the manuscript making the changes that you have recommended.
Reviewer 2 Report
The study compared the feeding outcomes of neonates who had heart surgery and were extubated in different time frames: in the OR, within 3 days or after 3 days. The study found no difference between the first two groups, but the last group had worse outcomes and longer hospital stay. The study suggested extubating within 3 days to improve feeding and reduce hospitalization. I read this paper with great interest and this paper is well organized and written. I have few comments and hope can improve the paper.
(1) This research is significant because it offers insights into the effect of extubating time on feeding outcomes after neonatal cardiac surgery. It provides evidence to suggest that early extubating improves feeding outcomes, which could help inform future clinical decisions and improve the care of newborns recovering from cardiac surgery. The potential issue of this research is that it is a single center study, which means that its results may not be applicable to other settings, populations, or contexts. Additionally, the results may be subject to bias due to the small sample size, as well as confounding factors that may not have been taken into account. Could you discuss this in the paper?
(2) I am thinking probably authors could combine Figure1 and 2 into one figure. That is, one figure has two panels, left and right.
(3) Missing data is a common issue in retrospective studies, which are those studies that look back on past events, such as medical records. This issue can arise if data was not collected or recorded correctly, or if it was lost or destroyed over time. I noticed that the proportion of missing data of this study is around 10%. Have you applied any methods to address this issue? Or may be authors could discuss the impact of missing data. Since it may bias the analysis results and lead to a wrong conclusion.
Author Response
Thank you for your careful review of our study and for your comments. Certainly single-center studies are full of potential limitations and biases. We have added language to the manuscript further exploring this as well as the missing data problem. We have also combine figures as suggested.
Round 2
Reviewer 2 Report
The revised manuscript looks ok. How about the thrid comment? I do not see any response to address that. Am I missing something?
Author Response
My apologies for not being clear. The only area where we had missing data was at 1 year follow-up. We chose to include these patients as we had data for the primary outcome and to be transparent about the missing data at the 1 year outcome, but we did not employee any special statistical methods. We have added a comment in the limitations pointing out this missing data and potential for bias in assessing outcome at 1 year.